# Nutrient Resorption and Stoichiometric Characteristics of Wuyi Rock Tea Cultivars

Dehuang Zhu [1,*], Suhong Peng [2], Wenzhen Liu [1], Shengjie Yu [1] and Dafeng Hui [3,*]

1 Fujian Provincial Key Laboratory for Eco-Industrial Green Technology, College of Ecology and Resources Engineering, Wuyi University, Nanping 354300, China
2 Department of Chemistry, College of Chemistry and Bioengineering, Yichun University, Yichun 336000, China
3 Department of Biological Sciences, Tennessee State University, Nashville, TN 37209, USA
* Correspondence: zhudh5@mail2.sysu.edu.cn (D.Z.); dhui@tnstate.edu (D.H.); Tel.: +86-158-8068-6123 (D.Z.); +1-615-963-5777 (D.H.)

**Abstract:** Nutrient resorption is an important strategy for plants to retain critical nutrients from senesced leaves and plays important roles in nutrient cycling and ecosystem productivity. As a main economic crop and soil and water conservation species, Wuyi Rock tea has been widely planted in Fujian Province, China. However, foliar nutrient resorptions of Wuyi Rock tea cultivars have not been well quantified. In this study, three Wuyi Rock tea cultivars (*Wuyi Jingui*, *Wuyi Rougui*, and *Wuyi Shuixian*) were selected in the Wuyishan National Soil and Water Conservation, Science and Technology Demonstration Park. Resorption efficiencies of nitrogen (NRE), phosphorus (PRE), and potassium (KRE) along with their stoichiometric characteristics were determined. PRE of the three tea cultivars was significantly higher than KRE and NRE, indicating that tea cultivars were P limited due to low P availability for the tea growth. With the exception of *Wuyi Rougui*, leaf N and P contents of the other two cultivars (*Wuyi Jingui* and *Wuyi Shuixian*) had strong homeostasis under the changing soil environments. Leaf thickness and specific leaf area were positively and significantly correlated with KRE, and total chlorophyll concentration was positively correlated with NRE, indicating that leaf functional traits can be used as indicators for nutrient resorption status. Wuyi Rock tea cultivars had strong adaptabilities to the environments and had high carbon sequestration capabilities; thus, they and could be introduced into nutrient-poor mountainous areas for both economic benefits and soil and water conservation.

**Keywords:** nutrient use efficiency; nutrient resorption; ecological stoichiometry; Wuyi Rock tea





## 1. Introduction

Nutrient resorption is an important strategy for improving nutrient utilization to facilitate plant growth and reduce plants relying on the soil nutrient status, especially in nutrient-poor environments [1,2]. Many factors influence plant nutrient resorption such as plant species, litter quality, and element characteristics [3,4]. Plant nutrient utilization and resorption status could influence ecosystem productivity and nutrient cycling processes in terrestrial ecosystems [2,5]. Therefore, the study of nutrient resorption could improve our understanding of the ecosystem nutrient limitations, nutrient cycling, and mechanisms of plant–soil feedback [6,7].

Different species have different resorption efficiencies. At the global scale, resorption efficiencies of nitrogen (NRE), phosphorus (PRE), and potassium (KRE) are higher than other nutrient elements [2]. Yuan and Chen (2009) reported that nutrient resorption efficiencies of the trees were lower compared to the shrubs, and deciduous trees had higher NRE compare to evergreen species [8]. Differences in nutrient resorption efficiencies could be attributed to both biotic (e.g., biological characteristics and leaf traits) and abiotic factors (e.g., climate conditions and soil nutrients) [5,9]. Leaf traits such as litterfall quality play an important role in N and P cycling in the forest ecosystems, and litter decomposition directly

affects the soil nutrients [10]. However, the linkage between litter nutrients and nutrient resorption efficiencies is still poorly understood [11,12]. Different litter types also show different nutrient resorption efficiencies, and nutrient stoichiometries provide insights into plant nutrient status [11,13]. In addition, forest stands profoundly regulate nutrient resorption [14,15]. Overall, leaf and litter nutrients have different effects on the plant nutrient resorption [16,17]. The relationships between leaf traits and nutrient resorption are still unclear, and the mechanisms of nutrient resorption are also not well understood.

Ecological stoichiometry is the study of the allocation and relationship of key elements of organisms, such as C, N, and P, and it is of great significance for the understanding energy flow and nutrient cycling in ecosystems [18]. C, N, and P are key elements for plant growth, and the study of the stoichiometry is helpful to explain the relationship and distribution characteristics of nutrient elements and describe the adaptability of plants in response to environmental changes. Ecological stoichiometry has been extensively applied to study nutrient recycling and nutrient limitation patterns in the terrestrial and aquatic ecosystems [19,20], but it is less used in agricultural research, especially in tea plantations [21,22]. Overall, the study of stoichiometry in tea ecosystems is still limited, and there is especially a lack of research on stoichiometry for above- and below-ground nutrient flow processes.

Soil nutrients also influence plant nutrient resorption. Previous studies found that the plants growing in the fertile soil have higher nutrient resorption efficiencies than in the unfertile soil [23,24]. However, high phosphorus and potassium resorption efficiencies are found in old-growth tropical forests with nutrient-poor soils [25]. Previous studies also found that N and P resorption efficiencies depend on physical–chemical processes of soil fertility along altitudinal gradients, especially the microbial P mineralization process providing a source of inorganic P for plant growth [6]. The impacts of soil nutrients on nutrient resorption efficiency may vary with different ecosystems.

As an important economic crop, tea cultivars have been widely planted in southern China [26,27]; the area of tea plantations reached 3.21 million ha in China in 2020, with 0.22 million ha in Fujian Province [28]. Tea plants are widely planted in hilly mountainous areas due to soil and water conservation functions [29]. Currently, the studies of tea cultivars mainly focus on the genetic diversity, metabolomics, and nutrient demand and cycling [30–32]. For example, fertilization has increased tea production and changed soil stoichiometry [19,33,34]. Age and population density have been found to regulate nutrient stock and soil stoichiometry in the tea ecosystems [20,35], and environments affect nutrient cycling in tea plantations [36]. However, leaf and litter stoichiometry has not been well assessed and the mechanisms of nutrient resorption of the tea cultivars remains unclear.

In this study, we selected three widely planted Wuyi Rock tea cultivars (i.e., *Wuyi Jingui*, *Wuyi Rougui*, and *Wuyi Shuixian*) in southeast China, and analyzed leaf nutrient resorption efficiencies, soil nutrients, and leaf functional traits. We tested the following three hypotheses: (1) tea cultivars would resorb more N and K than P as the study area was P impoverished; (2) leaf nutrients of tea cultivars would show strong homeostasis in response to nutrient-poor environments; (3) leaf functional traits could be used as indicators for nutrient resorptions of Wuyi Rock tea cultivars due to the close relationship among them. This study could provide insights into nutrient cycling and guidelines for tea plantation management.

## 2. Materials and Methods

### 2.1. Study Sites

The study site was located in the Wuyishan National Soil and Water Conservation, Science and Technology Demonstration Park (118°0′3.6″–118°0′28.8″ E, 27°43′55.2″–27°44′6″ N) in Wuyishan City, Fujian Province, Southeast China. A total area of the park is 55.75 ha, including a popular science education site, a scientific research pilot site, and a soil and water conservation demonstration site. This region has a typical subtropical monsoon climate, and the mean annual temperature is 17.1–18.1 °C, the mean annual frost-free

period is about 269 d, the mean annual sunshine time is 1910.2 h, and the mean annual precipitation is 1900 mm. The precipitation is mainly concentrated in spring and summer. The soil types are dominated by yellow soil and red soil. The site of water and soil erosion control in the demonstration park is 17.15 ha, including a tea park area of 7.75 ha. The main tea cultivars included *Wuyi Rougui*, *Wuyi Shuxian*, *Wuyi Jiulongpao*, *Wuyi Jingui*, and other high-quality *Camellia sinensis*. Tea trees were planted in rows, which facilitated the growth of tea trees and the picking of their leaves.

### 2.2. Experimental Design

At the soil and water conservation demonstration site, three representative Wuyi rock tea cultivars were selected (*Wuyi Jingui*-JG, *Wuyi Rougui*-RG, and *Wuyi Shuixian*-SX). They were planted in March 2019. A completed randomized experimental design with three replications for each tea cultivar was used. Three 1 m × 10 m plots with similar field conditions were randomly set for each tea cultivar, with a total of nine plots (90 m$^2$) for three tea cultivars. The height of each tea cultivar, plot coordinates, and other geographic information were recorded (Figure S1).

### 2.3. Sample Collection and Measurement

In July 2021, we randomly selected five tea trees in each plot, and 50 intact and fresh inner-canopy leaves were collected for each tea tree in four exposures: south-, east-, north-, and west-facing in each plot. A total of 15 tea trees for each tea cultivar, with a total of 45 leaf samples (15 samples × 3 tea cultivars) were collected. Simultaneously, we also collected five litter samples under the same tea tree in each plot, where litter had not been decomposed and bitten by insects, and a total of 45 litter samples. Fifteen trees for each tea cultivar were measured for total chlorophyll using a chlorophyll content meter (STYS-3, SINTEK, Shanghai, China) in the field. Meanwhile, five soil samples were collected at the 0–20 cm depth with a soil auger in each plot under each tea tree, and a total of 45 soil samples were sampled, brought back to the laboratory, and air-dried. All leaves, litter, and soil samples were ground with an agate mortar and then passed through 2 mm sieves. The fifteen sieved samples for each component of each tea cultivar for a total of 45 samples for leaves, litter, and soil were packed in polythene bags for elemental determination.

We chose five similar sized leaves for measuring specific leaf area (SLA), leaf thickness (LT), and leaf dry matter content (LDMC). SLA was calculated as the ratio of leaf area to the oven-dried leaf biomass. Leaf area was measured by a scanner (EPSONV600, Seiko Epson Corporation, Suwa City, Japan). The oven-dried leaf mass was determined by oven-dried leaves to a constant weight at 80 °C for 48 h [37]. LT was measured at the widest point of the main vein, with each leaf using a thickness meter (APL, Quzhou, China). LDMC was estimated by the oven-dried leaf biomass/the water-saturated fresh leaf biomass. Leaf C, litter C, and soil organic C were determined by the potassium dichromate (KCr$_2$O$_7$) method; leaf N, litter N, and soil total N were measured with the Kjeldahl method; leaf P, litter P, and soil total P were analyzed by digestion with H$_2$SO$_4$–H$_2$O$_2$ solution using molybdenum-antimony colorimetry; and leaf K, litter K, and soil total K were measured by flame photometry [38].

### 2.4. Data Analyses

Nutrient use efficiencies of N (NUE), P (PUE), and K (KUE) reflect the utilization and adaptation of plant to soil nutrients. Nutrient use efficiencies are used for analyzing the utilization of lower nutrients to increase plant biomass and characterize plant competition and survival strategies in poor soil environments [9]. The formula for *NuUE* was as follows [39]:

$$NuUE = \frac{M}{A_i} = \frac{M}{M \times C_i} = \frac{1}{C_i} \tag{1}$$

where *M* represents plant biomass (kg·ha$^{-1}$), $A_i$ represents nutrient storage of plant *i* (kg·ha$^{-1}$), $C_i$ represents nutrient (N, P, or K) content of different tea cultivars.

Nutrient resorption efficiencies of N (NRE), P (PRE), and K (KRE) are used for assessing the recovery of nutrients from senescing leaves and resorptions by nascent tissue. The calculation was based on Vergutz et al. (2012) [2]:

$$NuRE = \left(1 - \frac{\text{Nu}_{\text{senesced}}}{\text{Nu}_{\text{green}}} \times MLCF\right) \times 100\% \tag{2}$$

where $\text{Nu}_{\text{senesced}}$ and $\text{Nu}_{\text{green}}$ represent nutrient (N, P, or K) contents in senesced and green leaves of different tea cultivars, respectively. *MLCF* is the correction factor for mass loss, and the *MLCF* is 0.780 for evergreen forests.

Power function has been used to describe the allometric growth and link two attributes of the plant growth [20]. The equation can be converted to a linear function using a logarithmic transformation [40]:

$$\log(y) = b\log(x) + \log(a) \tag{3}$$

where *y* and *x* are the leaf nutrient contents of N, P, or K; *a* is the allometric exponent and *b* reflects the allometric constant. Standardized Major Axis (SMA) regression can be used with the package "smatr" in R 3.6.0 for the estimation of parameters *a* and *b*.

Plant homeostasis shows the ability of plants to maintain their chemical nutrient stability in response to environmental changes [41]. The slope of a linear function can be used to indicate homeostasis:

$$y = bx + a \tag{4}$$

where *y* and *x* are the leaf and soil N or P contents, respectively; *b* is the slope and *a* is the constant. When $b \neq 0$, it indicates that the nutrient of plant homeostasis is weak and changes with the soil. When $b = 0$, it indicates that the plant nutrients have absolute homeostasis.

Analysis of variance (ANOVA) and the Least Significant Difference (LSD) method were used for analyzing the differences of nutrients and their stoichiometric characteristics in different components among tea cultivars. The coefficient relationships between soil nutrients, traits, and nutrient resorption efficiencies were determined by Pearson correlation analysis. In order to evaluate the relationships among nutrient resorption efficiencies and leaf traits and soil nutrients, Principal Component Analysis (PCA) with standardized variables was applied, and the total variability of the first and second axis of PCA was more than 50%. All analyses and plots were performed using R 3.6.0 (R Development Core Team) and SigmaPlot 14.1.

## 3. Results

### 3.1. Leaf, Litter, and Soil C, N, P, and K Contents and Their Stoichiometry

Phosphorus content in the soil was the lowest among the three components in the three tea cultivars, and P content in the litter and soil of *Wuyi Jingui* was higher than those of *Wuyi Rougui* and *Wuyi Shuixian*. Of the three tea cultivars, K content showed in the order of soil > leaf > litter, and K content in the soil of *Wuyi Shuixian* was higher than that of *Wuyi Rougui* and *Wuyi Jingui*, but K content in the litter of *Wuyi Jingui* was significantly higher than that of *Wuyi Rougui* and *Wuyi Shuixian* (Table 1). The ratios of leaf C:K, leaf N:K, leaf P:K, soil C:P, and soil N:P of *Wuyi Rougui* were higher than those of *Wuyi Jingui* and *Wuyi Shuixian*. The ratios of litter C:P, litter N:P, litter C:K, and litter N:K of *Wuyi Jingui* were lower than *Wuyi Rougui* and *Wuyi Shuixian*. For each tea cultivar, the ratios of C:N in the soil and N:P in the leaf were the highest in three components (Table S1). For three tea species cultivars, the leaf nutrient contents (i.e., C, N, P and K) did not show obvious allometric relationships, and all the tea cultivars showed the similar trends (Table S2, Figure S2).

**Table 1.** The contents (mean $\pm$ standard error (SE)) of the organic carbon (C), total nitrogen (N), total phosphorus (P), and total potassium (K) in leaves, litter, and soil in the Wuyi Rock tea cultivars.

|  |  | C Content (mg/g) | N Content (mg/g) | P Content (mg/g) | K Content (mg/g) |
|---|---|---|---|---|---|
| *Wuyi Jingui* | Leaf | 533.5 $\pm$ 3.3 a | 22.1 $\pm$ 0.7 a | 2.1 $\pm$ 0.1 a | 12.8 $\pm$ 0.8 a |
|  | Litter | 537.0 $\pm$ 4.1 a | 18.6 $\pm$ 0.4 ab | 3.3 $\pm$ 0.1 a | 5.0 $\pm$ 0.4 a |
|  | Soil | 24.4 $\pm$ 1.3 a | 0.9 $\pm$ 0.1 a | 0.6 $\pm$ 0.1 a | 24.6 $\pm$ 1.1 b |
| *Wuyi Rougui* | Leaf | 552.1 $\pm$ 9.7 a | 24.7 $\pm$ 1.2 a | 2.3 $\pm$ 0.1 a | 9.5 $\pm$ 0.6 b |
|  | Litter | 511.2 $\pm$ 4.5 b | 18.1 $\pm$ 0.6 b | 1.8 $\pm$ 0.1 c | 2.9 $\pm$ 0.4 b |
|  | Soil | 12.9 $\pm$ 0.7 b | 0.4 $\pm$ 0.0 b | 0.1 $\pm$ 0.0 c | 16.2 $\pm$ 1.8 c |
| *Wuyi Shuixian* | Leaf | 542.9 $\pm$ 13.2 a | 25.6 $\pm$ 1.1 a | 2.4 $\pm$ 0.1 a | 14.4 $\pm$ 0.7 a |
|  | Litter | 530.5 $\pm$ 4.6 a | 20.8 $\pm$ 1.0 a | 2.5 $\pm$ 0.1 b | 1.9 $\pm$ 0.3 c |
|  | Soil | 11.0 $\pm$ 0.5 b | 0.3 $\pm$ 0.0 b | 0.2 $\pm$ 0.0 b | 54.6 $\pm$ 5.7 a |

Note: Different lower-case letters show significant differences in the same component of different tea cultivars.

### 3.2. Nutrient Use and Resorption Efficiencies of Different Tea Cultivars

In this study, different Wuyi Rock tea cultivars had different nutrient utilization and absorption efficiencies (Table 2). Potassium use efficiency of *Wuyi Rougui* (109.55 $\pm$ 4.94) was significantly higher than that of *Wuyi Jingui* (74.5 $\pm$ 1.4) and *Wuyi Shuixian* (72.0 $\pm$ 23.8). Nitrogen use efficiency of *Wuyi Jingui* was higher than *Wuyi Shuixian*. In each tea cultivar, nutrient use efficiencies showed in the order of phosphorus use efficiency > potassium use efficiency > nitrogen use efficiency. Nevertheless, PRE was the lowest in three nutrient resorption efficiencies of the Wuyi Rock tea cultivars. KRE of *Wuyi Shuixian* (89.7% $\pm$ 1.6) was significantly higher than that of *Wuyi Jingui* and *Wuyi Rougui*, and KRE was the highest in three nutrients of each tea cultivar. Interestingly, NRE was not significantly different among the three tea cultivars.

**Table 2.** The nutrient use efficiencies and nutrient resorption efficiencies of N, P, and K in different Wuyi Rock tea cultivars.

|  | NUE (%) | PUE (%) | KUE (%) |
|---|---|---|---|
| *Wuyi Jingui* | 45.9 (1.4) Ac | 471.3 (12.1) Aa | 74.5 (1.4) Bb |
| *Wuyi Rougui* | 41.6 (1.7) ABc | 438.8 (14.6) Ba | 109.6 (4.9) Ab |
| *Wuyi Shuixian* | 40.0 (1.7) Bc | 426.2 (16.6) Ba | 72.0 (3.8) Bb |
|  | **NRE (%)** | **PRE (%)** | **KRE (%)** |
| *Wuyi Jingui* | 33.2 (3.0) Ab | $-17.9$ (1.8) Cc | 72.0 (2.5) Ca |
| *Wuyi Rougui* | 41.1 (3.3) Ab | 39.3 (3.1) Ab | 76.2 (2.9) Ba |
| *Wuyi Shuixian* | 35.7 (3.2) Ab | 19.2 (1.5) Bc | 89.7 (1.6) Aa |

Note: Values are mean (standard error). NUE: nitrogen use efficiency. PUE: phosphorus use efficiency. KUE: potassium use efficiency. NRE: nitrogen resorption efficiency. PRE: phosphorus resorption efficiency. KRE: potassium resorption efficiency. Different capital letters of the same nutrient efficiencies and resorption efficiencies in the different tea cultivars show significant differences. Different lower-case letters of the different nutrient efficiencies and resorption efficiencies in the same tea cultivar show significant differences.

Different tea cultivars had different relationships among nutrient resorption efficiencies (Figure S3). For *Wuyi Jingui*, there was a negative linear relationship between NRE and KRE ($R^2 = 0.337$, $p = 0.023$). For *Wuyi Rougui* and *Wuyi Shuixian*, three nutrient resorption efficiencies had no significant linear relationship.

### 3.3. Relationships between Nutrient Resorption Efficiencies and Leaf and Litter Nutrients

Leaf and litter nutrients of different tea cultivars affected nutrient resorption efficiencies (Figure 1). Leaf and litter nutrients had significant linear relationships with nutrient resorption efficiencies. Leaf N content was positively correlated with NRE, and litter N had a significant negative relationship with NRE ($p < 0.05$). With the exception of negative relationships between litter P and PRE of *Wuyi Jingui* ($R^2 = 0.695$, $p < 0.05$) and *Wuyi Rougui* ($R^2 = 0.407$, $p < 0.05$), P content in leaf and litter showed no linear correlation with PRE.

In addition to leaf K, litter K had a significant relationship with KRE, and the correlation coefficient was above 0.7.

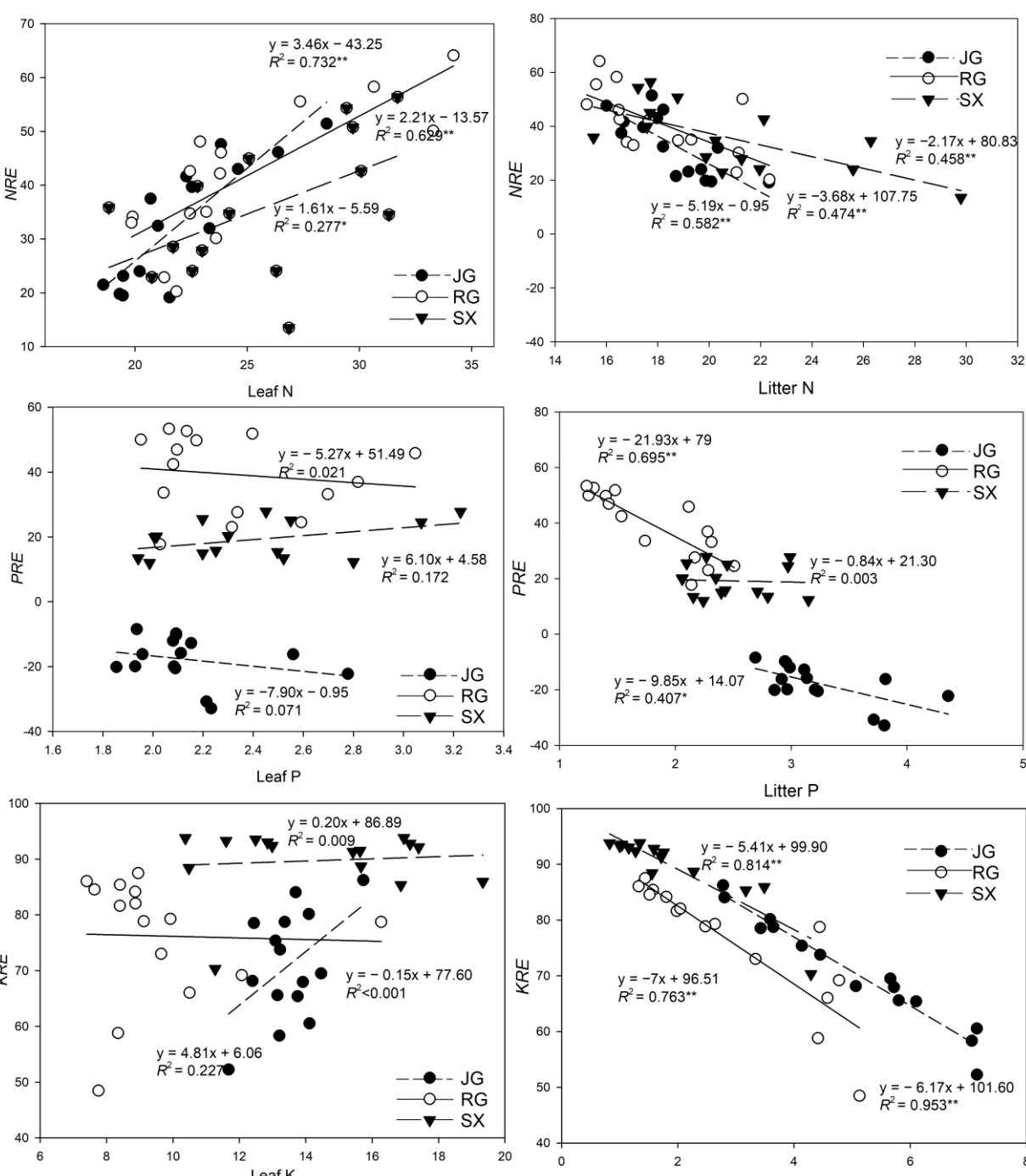

**Figure 1.** Relationships between nutrient resorption efficiencies and nutrient contents in the green and senesced leaves in the three Wuyi Rock tea cultivars. NRE: nitrogen resorption efficiency. PRE: phosphorus resorption efficiency. KRE: potassium resorption efficiency. No asterisk $p > 0.05$; * $p < 0.05$; ** $p < 0.01$.

Leaf N and P contents had different homeostasis with the changes in soil N and P content (Figure S4). For *Wuyi Rougui*, leaf N content had a negative correlation with soil P content, but leaf P content has a negative linear relationship with soil P ($p < 0.01$) (Figure S4c,d). For *Wuyi Jingui* and *Wuyi Shuixian*, there was no significant correlation

between leaf N and P contents as the linear regression slope and correlation coefficient were close to 0 (Figure S4a,b,e,f).

### 3.4. Relationships of Nutrient Resorption Efficiencies with Leaf Traits and Soil Nutrients

To further explore the relationships of nutrient resorption efficiencies with soil nutrients and leaf traits of three Wuyi Rock tea cultivars, PCA was performed. The total of the first and second axis of PCA explained 59.8%, 62.7%, 62.0%, and 55.0% of all three tea cultivars, and *Wuyi Jingui*, *Wuyi Rougui*, and *Wuyi Shuixian*, respectively (Figures 2 and 3). For all three tea cultivars, LT, SLA, soil C, and soil N aligned with the first axis but soil K and Chl aligned with the second axis, showing that PRE and NRE were correlated with soil C, N, and LT, and KRE was correlated with soil K and Chl.

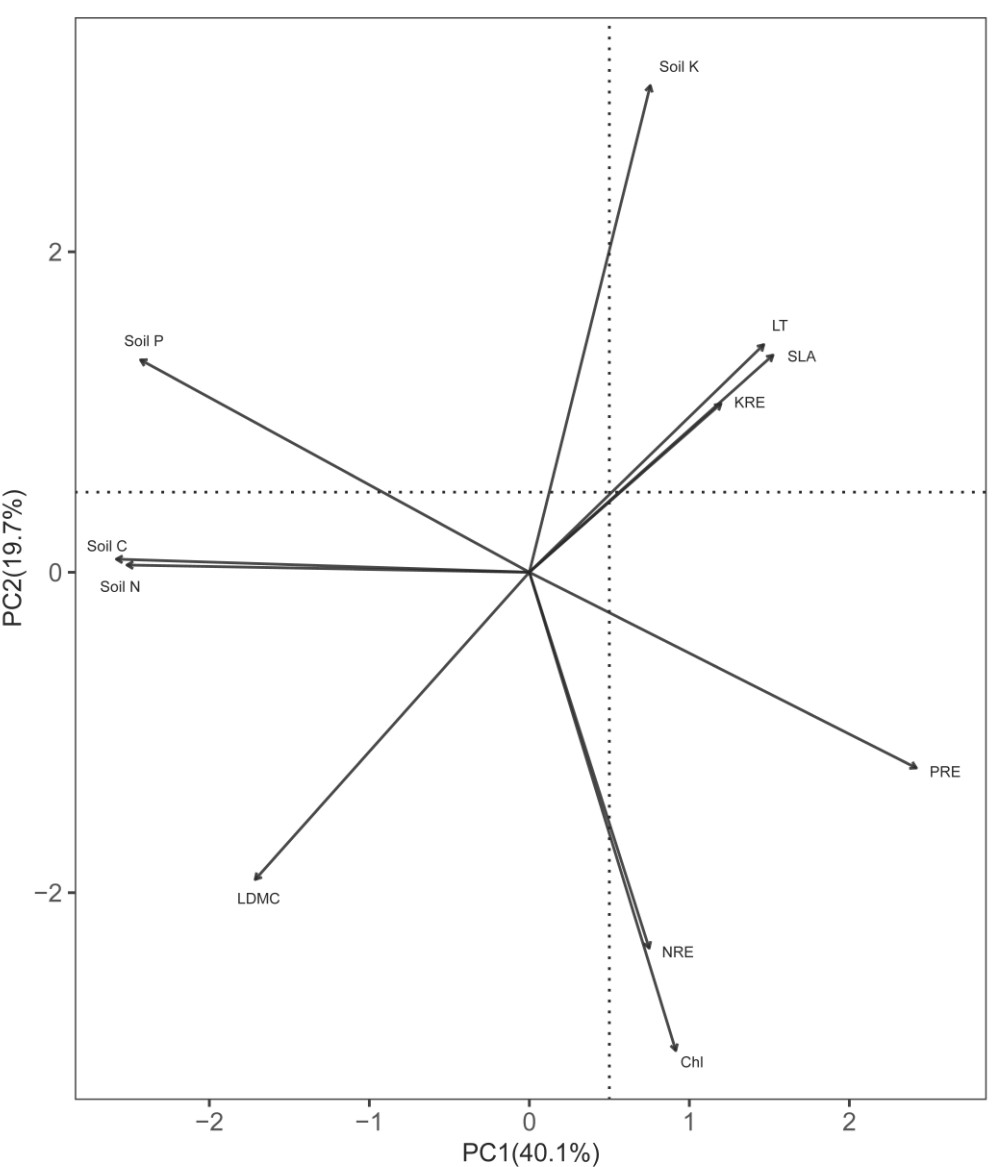

**Figure 2.** Principal Component Analysis (PCA) of nutrient resorption efficiencies, soil nutrients, and leaf traits for the total Wuyi Rock tea cultivars. SLA: specific leaf area. LT: leaf thickness. LDMC: leaf dry matter content. Chl: chlorophyll contents. NRE: nitrogen resorption efficiency. PRE: phosphorus resorption efficiency. KRE: potassium resorption efficiency. Soil N: soil total nitrogen content. Soil P: soil total phosphorus content. Soil K: soil total potassium content. Soil C: soil organic carbon.

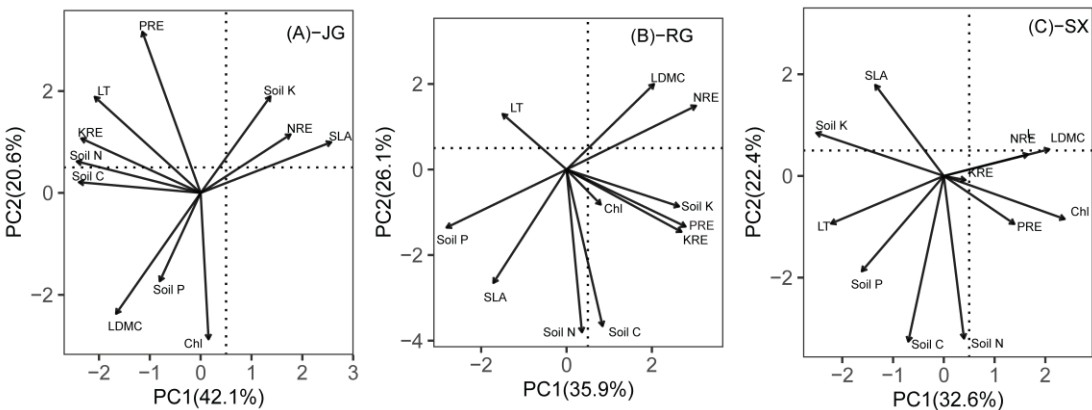

**Figure 3.** Principal Component Analysis (PCA) of nutrient resorption efficiencies, soil nutrients, and leaf traits in different Wuyi Rock tea cultivars. (**A**)-JG: *Wuyi Jingui*. (**B**)-RG: *Wuyi Rougui*. (**C**)-SX: *Wuyi Shuixian*. SLA: specific leaf area. LT: leaf thickness. LDMC: leaf dry matter content. Chl: chlorophyll contents. NRE: nitrogen resorption efficiency. PRE: phosphorus resorption efficiency. KRE: potassium resorption efficiency. Soil N: soil total nitrogen content. Soil P: soil total phosphorus content. Soil K: soil total potassium content. Soil C: soil organic carbon.

For *Wuyi Jingui*, leaf traits (e.g., LT and SLA) and soil nutrients (e.g., soil C and N) aligned with the first axis but LDMC and Chl aligned with the second axis, showing that KRE was correlated with soil C, N, and LT, and NRE was correlated with soil K and SLA. Nevertheless, soil P and K, LT, and LDMC generally aligned well with the first axis, SLA aligned with the second axis for *Wuyi Rougui*, showing that PRE and KRE were correlated with soil K and Chl. For *Wuyi Jingui*, the relationship between NRE and KRE was significantly negative (r = −0.58). SLA had a positive relationship with NRE, while it showed a negative correlation with KRE. LDMC was negatively correlated with NRE. LT had a positive relationship with PRE (r = 0.66) and KRE (r = 0.62). We also found that KRE was positively correlated with soil P and soil C (Table S3). For *Wuyi Rougui*, PRE was positively correlated with NRE (r = 0.60) and KRE (r = 0.69) (Table S4). NRE showed negative relationships with soil P and SLA, but a positive correlation with LDMC. LT was positively correlated with PRE and KRE. Meanwhile, PRE had a positive relationship with KRE (Table S4). For *Wuyi Shuixian*, there was no significant correlation among nutrient resorption efficiencies, with the exception for a negative relationship between NRE and soil K (Table S5).

## 4. Discussion

### 4.1. Stoichiometry of Wuyi Rock Tea Cultivars

The leaf N and P had no significant relationship for the three tea cultivars, indicating that N and P acquisitions were independent for each tea cultivar. There was also no correlation among other leaf nutrients of tea cultivars. This is due to the acquired nutrient by tea cultivars came from different sources such as atmospheric dust, soil fine particulate matter, and other organic debris. These results were different from previous studies of allometric relationships between the leaf N and P of plants due to their strong homeostasis response to the environment [20,42]. According to this study, the tea cultivars had a strong adaptation to the environment for the growth, and the nutrition of the tea cultivars showed high plasticity in response to different growing environments.

The relationships between plant growth and soil nutrition are often significant [43]. In this study, with the exception of C, N, and P in the leaf component, nutrients of litter and soil components were significantly different among different tea cultivars. Nutrients in the three components of each tea cultivar also had significant differences, and this was related to their biological characteristics. The leaf N and P nutrients of three tea cultivars were higher than those of the global plants (N: 20.1 mg/g, P:1.99 mg/g) [44] and China's

flora (N: 20.2 mg/g, P:1.5 mg/g) [45,46], while leaf C was lower than global plants and China's plants [44–46]. This was due to the weak acquisition of C resource and high N and P absorption and utilization efficiencies of tea cultivars. Leaf N:P is an important indicator of nutrient limitation for plants. When leaf N:P < 14, the plant growth is limited by N, when 14 < N:P < 16, the plant growth is restricted by N and P co-limitation, and when N:P > 16, P limitation was limited for the plant growth [47]. In this study, the ratio of N:P was lower than 14 for the three tea cultivars, indicating that growth of those tea cultivars was limited by N; this result reflected that the study area provided less available N for the Wuyi Rock tea growth. The ratios of leaf C:N and C:P of the three tea cultivars were lower than that of global plants [45,46], suggesting that the Rock tea cultivars had strong adaptation to infertile soil. Leaf K content was lower than the soil due to K in the litter being prone to return to the soil in the Rock tea cultivars.

Litter quality and quantity directly affects soil nutrients and regulates the nutrient cycling [48,49]. The contents of N and P in the litter for the three tea cultivars were higher than those of global plants [50], indicating that the tea cultivars had high nutrient utilization efficiencies due to the acquired nutrients from different sources, such as atmospheric dust and soil fine particulate matter. Litter C:N is an indicator of litter decomposition rate, and lower litter C:N had a higher litter decomposition rate due to the strength of the microbial activity and invertebrate digestion [51]. In this study, litter C:N in *Wuyi Shuixian* was lower than that of *Wuyi Rougui* and *Wuyi Jingui*, indicating that *Wuyi Shuixian* had a higher litter decomposition rate than *Wuyi Rougui* and *Wuyi Jingui*. Previous studies also reported that litter N and P are completely absorbed by leaves when litter N < 7 mg/g and P < 0.5 mg/g, but the litter N and P are not completed absorbed when litter N > 10 mg/g and P > 0.8 mg/g [10]. Three tea cultivars showed that N and P were not fully absorbed in this study, which were consistent with the lower nitrogen resorption efficiency and phosphorus resorption efficiency of three tea cultivars.

Soil nutrients are the main nutrient source for plant growth. The contents of C and N in *Wuyi Jingui* and *Wuyi Rougui* were higher than *Wuyi Shuixian*, indicating that *Wuyi Jingui* and *Wuyi Rougui* had strong C storage capacity. With the exception of soil organic C, the understory soil nutrients of the Wuyi Rock tea showed nutrient deficiency, especially P content. This was due to the Wuyi Rock tea cultivars having strong adaptation to barrenness, and they were used for soil and water conservation in the mountainous and hilly areas [29]. With the exception of *Wuyi Rougui* (416:10:1), the ratios C:N:P in the soil for *Wuyi Jingui* (111:3:1) and *Wuyi Shuixian* (124:3:1) were lower than global (186:13:1) [52] and Chinese soil (134:9:1) [53]. This indicated that the soil of *Wuyi Rougui* had a lower P resource for tea growth and *Wuyi Rougui* is more suitable for planting in P-deficiency areas. Conversely, Wuyi Rock tea cultivars showed a strong adaptation to the environment and high ecological benefits [26].

### 4.2. Nutrient Resorption Efficiencies of Different Tea Cultivars

Different Wuyi Rock tea cultivars had different nutrient resorption efficiencies. With the exception of KRE, the values of NRE and PRE of the Wuyi Rock tea were lower than those of global plants [2,54], due to different plant growth forms. A relatively low nutrient resorption in Wuyi Rock tea cultivars under a nutrient-limited area was in agreement with the leaf economic spectrum [55]. NRE and PRE of this study were also lower than trees and shrubs at the global scale [8]. Interesting, we found that the PRE of three tea cultivars had significantly lower values than NRE and KRE, indicating that P cycling was slow in Wuyi Rock tea ecosystem, which could reduce dependence on the P resource. Increased utilization of N and K, especially N, could alleviate the effect of P deficiency on the growth of tea tree. These results supported our first hypothesis. The information of NRE and PRE was helpful to understand nutrient-use strategies and nutrient limitation of plant [56,57]. KRE (>70%) was higher in this study than NRE and PRE, as more K was resorbed from senesced leaves to mature leaves than was retained in the senesced leaves,

further confirming that K was resorbed preferentially during the process of regeneration of senesced and new leaves [58,59].

The scaling exponents of NRE versus PRE of the Wuyi Rock tea cultivars were smaller than 1, which was consistent with previous reports [45,60]. Compared to P, most of leaf N is invested in Rubisco, and could be re-mobilized from the senesced leaf [61]. Foliar nutrient status has been assessed to understand the nutrient resorption patterns [62]. In this study, there was a positive relationship between the green leaf N and NRE, suggesting that tea cultivars tended to resorb more N nutrient for offsetting the P deficient. These results were consistent with the previous studies [54,57]. Nutrient resorption efficiencies had significant negative relationships with senesced leaf nutrients in this study, indicating that more nutrients in the senesced leaves were reabsorbed when nutrient resorption efficiencies were higher [17]. These findings confirmed that environmental and biological factors drive the nutrient resorption plasticity [5].

Different tea cultivars had different nutrient resorption efficiencies. All three tea cultivars had low PRE and high KRE in this study. The lower PRE was attributable to lower C:P in the soil, indicating that the study site was P deficiency. These findings were in accordance with the results of Chen and Chen (2021) [14]. The low P and high K in soil, mainly caused by tea tree litter input, could increase potassium use efficiency and reduce phosphorus use efficiency for tea cultivars. Moreover, the low PRE of Wuyi Rock tea enhanced the P nutrient return to soil during the litter decomposition [24]. This study site had low P nutrient in the soil, resulting in positive effects on P uptake. In addition, more N remained in the senesced leaves of the tea cultivars, highlighting that the tea ecosystem accelerated N cycling and reduced dependence on the P resource for the tea growth in the P-poor site. These findings were different from some previous studies that reported improved P cycling for plant growth [9,14]. Further, the high KRE for the three tea cultivars was attributable to high K in plants and soil systems.

### 4.3. Response of Nutrient Homeostasis and Nutrient Utilization Strategies

Homeostasis theory holds that organisms maintain the relative stability of its elements to adapt to the changes in the external environments [63,64]. In this study, leaf N and P contents of *Wuyi Jingui* and *Wuyi Shuixian* had strong homeostasis, indicating that the leaf N and P nutrients were less vulnerable to soil nutrient status. Nevertheless, leaf N and P contents of *Wuyi Rougui* showed low homeostasis, indicating that leaf N and P contents changed with soil P contents. These results partially supported the second hypothesis. Compared to leaf N of *Wuyi Rougui*, leaf P had stronger homeostasis, this finding was inconsistent with the results of Yu et al. (2010) due to N limitation in that study [64]. Our results, to a certain extent, also verified "the Stability of Limiting Elements Hypothesis" [65], that is, limiting elements remained relatively stability in response to the changes in the external environments. These results also indicated *Wuyi Jingui* and *Wuyi Shuixian* could be planted in N and P deficiency areas, especially in mountain hill areas.

The nutrient resorption efficiencies of different tea cultivars had different relationships with leaf traits. For *Wuyi Jingui* and *Wuyi Rougui*, leaf traits mainly affected PRE and KRE, but NRE was closely related to leaf traits for *Wuyi Shuixian*. Our results indicated that leaf functional traits of tea cultivars might be a good indicator of NRE. This finding seemed to be in accordance with earlier studies [15,66]. Overall, our results supported the third hypothesis.

In this study, soil nutrients had no relationship with leaf N and P contents, but green leaf nutrients of tea cultivars directly regulated the nutrient resorption efficiencies. N, P, and K of the senesced green leaves of different tea cultivars had significant effects on the nutrient resorption efficiencies, but nutrient resorption efficiencies were less affected by the nutrients of green leaves. The PRE and KRE of *Wuyi Rougui* and *Wuyi Shuixian* were higher than that of *Wuyi Jingui*, indicating the strong acquisition and resorption of P and K in the soil erosion area. In the management of tea trees, *Wuyi Rougui* and *Wuyi Shuixian* were more suited to planting in the P and K deficient area than other Wuyi Rock tea cultivars.

Therefore, *Wuyi Rougui* and *Wuyi Shuixian* are suitable species for restoration in soil and water conservation areas.

## 5. Conclusions

In this study, N and P were not fully resorbed and retained by the senesced leaves of the Wuyi Rock tea cultivars, but K was resorbed resulting in high-K use efficiency. Leaf traits such as leaf thickness and specific leaf area might be useful indicators of nutrient resorption for the Wuyi Rock tea cultivars. We also demonstrated that three tea cultivars at the site were more P limited than N, and improving N and K utilization and uptake could offset the deficiency of P. *Wuyi Shuixian,* which had the highest adaptation to the environment compared to two other tea cultivars, and it could grow in complex and changing environments. Our results indicated that *Wuyi Rougui and Wuyi Shuixian* are more suitable than *Wuyi shixian* to grow in a P-deficiency environment. N and P fertilizers can be added for the tea cultivars' management in the future.

**Supplementary Materials:** The following supporting information can be downloaded at: https://www.mdpi.com/article/10.3390/f14040675/s1, Figure S1: Schematic diagram of the study site; Figure S2: Bivariate plots among the leaf carbon (C), nitrogen (N), phosphorus (P), and potassium (K) in the tea species; Figure S3: The correlations among of nitrogen resorption efficiencies (NRE), phosphorus resorption efficiencies (PRE) and potassium resorption efficiencies (KRE) in three Wuyi Rock tea cultivars; Figure S4: Relationships between leaf nitrogen and phosphorus contents with respective soil nitrogen and phosphorus contents in three Wuyi Rock tea cultivars; Table S1: The contents (Mean ± SE) of stoichiometric for the organic carbon (C), total nitrogen (N), total phosphorus (P) and total potassium (K) in leaves, litter, and soil in the Wuyi Rock tea species; Table S2: Summary of reduced major axis regression analyses (slope and y-intercept, allometric index) of leaf nutrients stoichiometry for three Wuyi Rock tea cultivars; Table S3: The correlation analysis between nutrient resorption efficiencies, leaf traits, and soil nutrients for *Wuyi Jingui*; Table S4: The correlation analysis between nutrient resorption efficiencies, leaf traits, and soil nutrients for *Wuyi Rougui*; Table S5: The correlation analysis between nutrient resorption efficiencies, leaf traits, and soil nutrients for *Wuyi Shuixian*.

**Author Contributions:** Conceptualization, D.Z., S.P. and D.H.; funding acquisition, D.Z. and D.H. writing—reviewing and editing, D.Z. and D.H.; field investigation, Experimental and data analysis, D.Z., W.L. and S.Y. All authors commented on the previous versions of the manuscript. All authors have read and agreed to the published version of the manuscript.

**Funding:** D.Z. was supported by the Natural Science Foundation of Fujian Province, China (Grant number 2021J05248), Science Foundation Project for Talented New Faculty at Wuyi University (grant number YJ202103). D.H. was supported by the US NSF and USDA projects.

**Data Availability Statement:** Not applicable.

**Acknowledgments:** We thank Liu Yinni, Guan Yanmeng, Deng Xinyin, Li Liting, and other people for the field and laboratory assistance.

**Conflicts of Interest:** The authors have no conflict interests in this paper.

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
