# Peer review of "Nutrient Resorption and Stoichiometric Characteristics of Wuyi Rock Tea Cultivars"

_forests, doi:10.3390/f14040675_

Round 1

Reviewer 1 Report

The authors investigated nutrient uptake and ecological stoichiometry of the three tea varieties. Their aim was to determine the resorption efficiencies of N, P and K in relation to the elemental profile of leaves, litter and soil, as well as leaf thickness, leaf area, leaf mass and leaf chlorophyll content. The leaf N and P contents of two of the three tea cultivars showed strong homeostasis under different conditions. Leaf thickness and area were correlated with potassium uptake efficiency, and total chlorophyll concentration was correlated with nitrogen uptake efficiency, indicating that leaf nutrients and leaf characteristics can be used as indicators of nutrient uptake status. The study concludes that Wuyi Rock tea cultivars have strong environmental adaptability and high carbon sequestration capacity, and could be introduced into nutrient-poor mountainous areas for both economic benefits and soil and water conservation. Overall, the study is an interesting contribution to the understanding of plant ionomics and the role of plants in nutrient cycling. Since the manuscript doesn't contain any serious errors, I think it could be acceptable if revised.

Comments

I would recommend always using full words instead of abbreviated ones. Abbreviations make it difficult to understand the text because they are hard to remember and understand.

The authors refer to stoichiometry for the first time in the description of methods on line 173. An earlier explanation of what ecological stoichiometry and ionomics are would have made this study easier to understand. Can the authors provide a short paragraph in the introduction that describes the fundamentals of ecological stoichiometry and ionomics and the application of these frameworks to the topics addressed in this manuscript (conservation biology, agriculture, nutrient cycling, etc.)?

Line 133
I do not understand what the authors mean by specific leaf area in this study. Please define it.

Line 150
What does hm mean in (kg*hm-2) ?

Lines 247-258
What do the authors mean by “N and P were relatively independent” ? I don't understand this sentence

Lines 249-251
This sentence is strange. Can the authors elaborate and lighten it?

Lines 252-253
If the results of this study are different from those currently reported in the literature, the authors should offer a brief explanation of the results obtained.

Line 254
Citation is needed

Lines 273-275
Why do high N and P contents in litter suggest high nutrient utilisation efficiency? My understanding is that it should be the other way around. Please clarify this reasoning.

Reviewer 2 Report

I have no objections to this text. 

Question for line 140: the K has been measured by flame photometry; from the same digest as P? Hopefully, Cs ionization buffer was used.

The numbers in tables 1 and 2 should be cut to significant values, in order to reflect possible obtainable precisions

Do the results have any meaning for the farmer to chose a suitable variety of tea leaves?

Round 2

Reviewer 1 Report

Contrary to what is stated in the authors' response to the reviewers' comments, the authors did not provide a paragraph in the manuscript explaining what the ecological stoichiometry and ionomics they used actually are. I therefore again ask the authors to write a meaningful and comprehensive paragraph on the research framework used in the study to make this study easier to understand.

In the introductory section of a scientific manuscript, it is essential to explain the research framework that is used in the study. The introduction serves as a critical element in presenting a concise and understandable overview of the research problem being investigated. It is important to establish the purpose of the study and to emphasise the importance of the topic under investigation, which can be achieved by providing a compelling rationale for why the research question is worth investigating. To achieve this, authors should identify gaps in current knowledge or shortcomings in existing research, which will highlight the importance of the research question. By presenting the research framework, authors can provide readers with a better understanding of the theoretical foundations of the research, thereby increasing the overall credibility of the manuscript. In addition, providing a clear research framework can help readers better understand the methods used in the study and the relevance of the results obtained. Therefore, please explain to the reader what ecological stoichiometry/ionomics is and why this framework is particularly appropriate for addressing the research questions posed by the authors in this manuscript. What is valuable about this approach? How/why is it relevant to the problem under investigation? How does this approach, which focuses on inorganic chemistry, relate to the functioning of nature and, ultimately, how can it be applied to conservation? This contextual background helps to engage the reader and sets the stage for the following sections of the manuscript.

Round 3

Reviewer 1 Report

I have no comments.